# Household Sharing for Carbon and Energy Reductions: The Case of EU Countries

**Diana Ivanova * and Milena Büchs**

School of Earth and Environment, University of Leeds, Leeds LS2 9JT, UK; M.M.Buchs@leeds.ac.uk
* Correspondence: d.ivanova@leeds.ac.uk

**Abstract:** As households get smaller worldwide, the extent of sharing within households reduces, resulting in rising per capita energy use and greenhouse gas (GHG) emissions. This article examines for the first time the differences in household economies of scale across EU countries as a way to support reductions in energy use and GHG emissions, while considering differences in effects across consumption domains and urban-rural typology. A country-comparative analysis is important to facilitate the formulation of context-specific initiatives and policies for resource sharing. We find that one-person households are most carbon- and energy-intensive per capita with an EU average of 9.2 tCO$_2$eq/cap and 0.14 TJ/cap, and a total contribution of about 17% to the EU's carbon and energy use. Two-person households contribute about 31% to the EU carbon and energy footprint, while those of five or more members add about 9%. The average carbon and energy footprints of an EU household of five or more is about half that of a one-person average household, amounting to 4.6 tCO$_2$eq/cap and 0.07 TJ/cap. Household economies of scale vary substantially across consumption categories, urban-rural typology and EU countries. Substantial household economies of scale are noted for home energy, real estate services and miscellaneous services such as waste treatment and water supply; yet, some of the weakest household economies of scale occur in high carbon domains such as transport. Furthermore, Northern and Central European states are more likely to report strong household economies of scale—particularly in sparsely populated areas—compared to Southern and Eastern European countries. We discuss ways in which differences in household economies of scale may be linked to social, political and climatic conditions. We also provide policy recommendations for encouraging sharing within and between households as a contribution to climate change mitigation.

**Keywords:** household size; household economies of scale; carbon footprint; energy footprint; consumption; European Union; urban; rural; population density; climate change mitigation

## 1. Introduction

We need rapid and effective climate action to reduce global greenhouse gas (GHG) emissions and avoid catastrophic climate change. Annual emissions must decrease to close to half of their 2010 levels by 2030, and reach net-zero by 2050 to increase the probability of limiting temperature changes to 1.5 °C above preindustrial levels [1]. Yet there are some socio-demographic trends that may make it more difficult to achieve this.

One such trend is the shrinking of household sizes globally. Together with the rise in global emissions, the number of households has also been increasing, outpacing population growth. Several studies have shown that there is a strong link between household size and per capita energy use and GHG emissions [2–6] in both developed and developing countries [7]. When individuals live together, there are "economies of scale"—people tend to share appliances, tools and equipment, cook together and heat and cool common living spaces. These acts of sharing allow for the per capita energy use to diminish with rising household size. Thus, as households get smaller, the extent of sharing within

households reduces, while the per capita energy use and emissions rise. Some domains, such as energy consumption for heating, cooling and lighting, show substantially higher potential for household economies of scale [3,4,8] compared to others such as transport, clothing, and services [2,4,8]. There may also be social benefits associated with shared living and larger household sizes, as they tend to counteract trends of isolation and loneliness and build stronger communities [9,10]. Furthermore, recent research shows that members of grassroots initiatives, which may involve communal living such as eco-villages and Transition towns, manage to reconcile lower carbon footprints and less materialistic living with higher life satisfaction [11,12]. Recognizing the important role of household economies of scale and their social and environmental implications, researchers have advocated policies and initiatives that encourage larger households and sharing within and across households [13].

Yet, the majority of research evidence focusing on the role of household size for consumption-based energy and GHG emissions is restricted to single country studies. A notable exception is a comparative multivariate analysis of household energy requirements in Australia, Brazil, Denmark, India and Japan conducted by Lenzen and colleagues dating from 2006 [14]. There is a lack of up to date comparative studies between countries [5], examining these trends in a broader context and discussing the potential contextual differences across countries. An up-to-date country comparative perspective is important from a policy perspective: is advice on supporting sharing within and across household equally valid across all EU countries, or do these strategies need to vary and be adjusted to different contexts?

Furthermore, average household sizes differ between rural and urban areas and opportunities to share may also vary with urban-rural context [15,16]. Yet, studies that examine the interaction between household size and population density in a country-comparative setting are lacking. This article addresses this gap, analyzing the role of household size and its interaction with urban-rural typology across EU countries.

Our main finding is that household economies of scale vary substantially across consumption categories, urban and rural typology and EU countries. High household economies of scale are noted for home energy, real estate services, and miscellaneous services such as waste treatment and water supply; yet, some of the weakest household economies of scale occur in high carbon domains such as transport. Furthermore, Northern and Central European states are more likely to report strong household economies of scale—particularly in sparsely populated areas—compared to Southern and Eastern European countries. We discuss possible reasons for these patterns, as well as policy strategies to encourage sharing within and between households to contribute to climate change mitigation.

## 1.1. Cross-Country Differences in Household Economies of Scale

There may be various factors explaining the potential differences in the household economies of scale across EU countries. Some of these are related to the distribution of household size and composition. Adding another member to a household is likely to reduce per capita energy use and carbon footprints at a decreasing rate with rising household size; that is, increasing the household size from one to two members may drastically reduce home energy use and the associated carbon footprint, while a change from three to four members has been shown to produce a smaller effect on average [2]. Furthermore, the household composition—e.g., the age and the gender of the new household member—may also play an important role [3,5].

Several social, political and cultural factors are also likely to influence the effect that an additional household member has on energy and carbon footprints. Widely reported long-term changes include decreasing social trust, concern for others, conformity and religiosity, and increasing individualism, gender egalitarianism, materialistic and extrinsic values [17,18], all of which may have implications for household dynamics and sharing practices. Yet, following the global financial crisis, more recent changes in values towards greater importance of conservation (security, tradition) and concerns for close others (benevolence) have been noted in Europe [17]. As countries with extensive social nets report lower value changes following the financial crisis [17], we discuss social welfare systems as an important country-specific factor that influences the potential for sharing within and between households. Welfare

regimes that promote individual independence, female participation in the labor force, and countries with higher levels of secularization [19] may stand out with lower household sizes, which may also affect the potential for household economies of scale. Differences in consumption patterns across countries, stemming from differences in culture, social norms, geography and climate, infrastructural and institutional contexts, may also explain some of the variation in household economies of scale.

While we cannot test these theories directly in our analysis, they offer potential explanations for the country clustering of household economies of scale in our analysis.

*1.2. Interaction between Household Size and Population Density*

Urban areas are associated with high population and employment densities, compact and mixed land uses, and high degrees of connectivity and accessibility [20–22]; as such they have higher potential for collaborative consumption and sharing of resources between households, and more efficient uses of infrastructure [5], the so-called "compact" or "density effect" hypothesis [16]. This is because urban areas with narrower streets and smaller city blocks, compact and connected design, pleasant and safe urban space and mixed land uses generally reduce travel distance and promote active travel (walking and biking) and public transport [5,20,22]. Furthermore, urban dwellings are associated with smaller sizes, a higher proportion of apartments and multi-family houses and the presence of district heating, which are overall less carbon and energy intense per unit of area [3,22]. While there is strong evidence for this density effect on per capita carbon and energy footprints in the European context, this is largely compensated by higher income levels in cities [23]. Urban cores are generally preferred by more affluent and younger adults with greater consumption opportunities and smaller household sizes (and hence higher per capita carbon and energy footprints), while suburban areas benefit from larger household sizes and economies-of-scale effects at the household level [15,16]. This clearly complicates the established view that dense urban environments are more sustainable [5].

Furthermore, household economies of scale are likely to differ between rural and urban areas [15,16]. A recent study from the USA found household economies of scale to be about twice as large in rural compared to urban contexts (up to 8% reduction in per capita carbon emissions when adding an adult in rural contexts compared to 3% reduction in dense urban contexts) [24]. Lower household economies of scale in urban areas have also been found in a European context [25]. An explanation for this trend is that both household and urban economies of scale "are driven by proximity and realized through sharing" [24]. Adding a member in a rural detached house will bring about higher savings through sharing walls, living space and heating and cooling, compared to adding a member in a shared apartment building, where walls are already shared between more households, living space is smaller and common district heating may be present. Urban context is associated with proximity between households and thus higher potential to share resources outside of the household, which may in turn partially offset the household size effect. We explore differences in the household economies of scale between urban and rural context through an interaction term between household size and population density in the model.

*1.3. This Study*

In this article, we calculate the total and the average per capita EU carbon and energy footprints for various household sizes. We examine the inter-country differences in household economies of scale across 26 EU countries as a way to uncover sharing opportunities and support reductions in energy use and GHG emissions. This analysis considers differences in effects across consumption domains, as well as between rural and urban areas.

Prior studies generally focus on a single country, while a comparative perspective is lacking. A comparative perspective allows for a more robust discussion of the potential energy and GHG emission cuts that could be achieved through within- and between-household sharing—and may help formulate context-specific initiatives and policies for resource sharing on a regional and country level.

Studies usually focus on either carbon or energy. In this study, we examine both in order to enable a wider comparison.

## 2. Data and Methods

### 2.1. Databases

The Household Budget Surveys (HBS), harmonized and disseminated through Eurostat, collect information about household consumption expenditure across EU countries. This study utilized data from 2010, which is the latest available. Price coefficients were used to adjust household expenditure to the reference year of 2010 and EUR/purchasing power standard (PPS) units, thus accounting for price differences across countries and time (for the countries, which collected expenditure in a different year) [26]. A detailed overview of the HBS accuracy (sampling and non-sampling errors), timeliness, comparability and representativeness is provided elsewhere [26]. We transformed household expenditure into per capita expenditure and proceeded with carbon and energy footprint calculations.

We calculated annual carbon and energy footprints on the household level, utilizing the multiregional input-output database EXIOBASE (version 3.7) [27]. We applied the Global Warming Potential (GWP100 [28]) metric to convert various GHGs (carbon dioxide, methane, nitrous oxide and sulphur hexafluoride) to kilograms of $CO_2$-equivalents per year ($kgCO_2eq$). Annual energy use was calculated using the net energy extension measures in terajoules (TJ). There is no double counting with regards to the conversion from primary sources (derived directly from nature, e.g., coal) into secondary sources (coal-generated electricity, for instance) [29]. In this paper, we used the terms "carbon footprints" and "GHG emissions", as well as "energy footprints" and "energy use" interchangeably. We expected that the two environmental indicators would depict similar trends in terms of the effect of household size, as the majority of GHG emissions are related to energy use (e.g., burning of fossil fuels).

The EXIOBASE database covers high sectoral detail (200 products), 49 countries (including all EU countries) and rest-of-the-world regions, and a wide range of environmental and social satellite accounts [27,30]. We matched the HBSs household expenditure in 2010 with the environmental and economic structure in EXIOBASE for the same year. For a detailed overview of the harmonization steps between consumption from HBSs and the environmental intensities from EXIOBASE, see SM1 and elsewhere [4,31].

### 2.2. The Model

In order to examine inter-country differences in household size effects, we performed the regression analysis for each EU country *c* separately (see SM4 for a robustness check through a model including all of the countries). We also performed the analysis on EU level. We applied the household weights disseminated by Eurostat. The analysis is conducted on a per capita level for each household *i*, with the following specified model:

$$ln\big(\widehat{ENVF_{ct}}\big)$$
$$= \beta_{c0} + \beta_{c1}(LNINCOME_{ci}) + \beta_{c2}(HHSIZE_{ci}) + \beta_{c3}(DENSE_{ci}) + \beta_{c4}(INTERMEDIATE_{ci}) + \beta_{c5}(HHSIZE_{ci}$$
$$\times DENSE_{ci}) + \beta_{c6}(REGION_{ci}) + \epsilon_{ci}$$

*ENVF* stands for the estimated environmental footprint, namely the annual carbon or energy footprint per capita measured in $kgCO_2eq$ and TJ, respectively, in logarithmic form. The log-transformation was done to achieve normally distributed regression residuals, which previously had a positively skewed distribution.

*LNINCOME* measures the role of net disposable household income [32] (not equivalized) for the environmental footprint. The income coefficient can be interpreted as income elasticity as both the dependent and independent variables are measured in logarithmic form. As the Italian HBS does not

include the income variable used for other countries, we employed the logarithm of total expenditure instead as an independent variable, similar to other studies [14,24].

*HHSIZE* presents the number of household members. The term *household* refers to people with a common use of an address, usually sharing space and practices [9]. In the HBSs, sharing common accommodation and expenses was also central to the household definition.

The dummy variables for population density (*DENSE* and *INTERMEDIATE*) utilize the Eurostat's measure of the degree of urbanization [33], based on Local Administrative Units level 2 (LAU2). LAU are low level administrative divisions below that of a province, region or state [34], where LAU2 is the lowest consisting of municipalities or equivalent units in the 28 EU Member States (formerly NUTS 5 level) [35]. The degree of urbanization defined by Eurostat classifies LAU2 into sparsely, intermediate and densely populated areas, using as a criterion the geographical contiguity in combination with the population density in the different types of areas [33]. A map of the degree of urbanization in 2011 for all of the EU and a detailed explanation of the undertaken steps for the LAU2 classification can be found elsewhere [33]. In this article, variable *DENSE* takes the value of one for households that live in areas with at least 500 inhabitants/km$^2$, and zero otherwise (cities). *INTERMEDIATE* takes the value of one for households that live in areas between 100 and 499 inhabitants/km$^2$, and zero otherwise (towns and suburbs). The base category *SPARSE* is associated with rural or sparsely populated areas with less than 100 inhabitants/km$^2$ according to the HBS classification.

Similar to a prior study [24], we added an interaction term between household size and population density (*HHSIZE×DENSE*) in order to explore the potential variability in household economies of scale by urban-rural typology.

We also included spatial controls—a set of regional dummy variables (*REGION*)—aiming to account for regional differences such as technological (e.g., energy efficiency or infrastructure, type of dominant industries) as well as geographical and climatic context [4] (see SM1 for an overview of all regions). The regional distribution is the first-level NUTS of the EU for most countries.

Prior work has discussed the selected variables in the model as key socio-demographic, economic and geographical determinants of environmental footprints [4,5,24]. While additional factors such as dwelling size and type, vehicle ownership, energy sources and prices [3,21] among others are important, the HBSs do not collect such data. We also did not explore the role of household composition, while prior studies found education, gender and age to have small and mixed effects [2–5,36]. For example, females have been found to have lower carbon footprints associated with transport and food, and higher energy use at home [3,36]. Single parent households (mostly headed by women) were found to be more likely to experience fuel and energy poverty [37]. Age has been found to be positively associated with energy use [3,38], although this effect may slow down or even change direction when people reach their later years [2,36]. Education and social status may also redesign preferences towards more or less emission- and energy-intensive consumption [2,4,39].

We estimated the regression model based on household surveys from 25 EU countries (excluding Sweden and the Netherlands due to lack of consumption data and Romania due to lack of population density), with a total sample of 243,911 observations.

## 2.3. Limitations

Our analysis was affected by limitations regarding the representativeness, harmonization and measurement errors of the HBSs. A detailed account of these limitations [26] and their implications for the carbon and energy footprint calculations can be found elsewhere [3,31]. There may be higher sampling error and inflated variation associated with infrequent purchases [26], for instance second homes [40], personal vehicles, flights or furniture, and their associated environmental impacts.

There are some limitations regarding the environmental impact assessment. EXIOBASE offers details of 200 products and services across 44 countries and five rest-of-the-world regions, and can thus only distinguish the country-level carbon and energy intensities of largely heterogeneous product groups. Particularly in the context of household dynamics, the product detail was insufficient to

distinguish between consumption of items that are more likely to be shared within and between households (e.g., use of shared appliances vs. individual equipment). Difficulties in allocating land use change emissions to specific economic activities have been previously recognized [41,42].

Some products and services may also be purchased directly by households in some countries but are provided through governmental spending in others. Focusing solely on household expenditure may thus result in substantial variation in terms of spending on health, social work, education and transport services, disregarding impacts associated with public provision, which affects comparative analysis [43]. As a result, our analysis may not capture well country differences in the between-household sharing opportunities through the provision of public infrastructure.

Furthermore, as household carbon and energy footprints are based on monetary expenditure, there are limitations due to potential price differences within products. Therefore, we likely overestimated the environmental impact of expensive products (wealthier individuals) and underestimated the impact of cheap products (and less wealthy individuals) [44]. In addition, we could not examine the effect of "green consumerism" [16] on carbon and energy intensities, e.g., buying a fuel-efficient car, opting for a green energy provider or a more expensive but energy efficient dwelling. Larger households may also be more likely to purchase items in bulk and, thus, pay lower prices per item. Prior work discusses the limitations associated with the monetary-based approach [2,31,44].

The HBS uses household size or type in the stratification criteria for most countries in order to make the survey sampling more accurate [26,32]. Yet, there may be an under-representation of less common household types such as intentional communities (e.g., eco-villages, co-housing). All collective households such as elderly homes, boarding schools and others, where individual spending cannot be distinguished from collective spending, have been excluded from the HBSs [26].

Furthermore, the population density variable and interaction effect are based on the LAU2 classification and as such it can only capture potential consumption and footprint differences between cities, towns and suburbs and rural areas. We cannot capture differences in the between-household sharing potential and opportunities on dwelling-, close community- or neighborhood levels.

## 3. Results

### 3.1. Descriptive Statistics and Bi-Variate Regressions

#### 3.1.1. Household Size, Carbon and Energy Footprints

In per capita terms, one-person households have the highest average carbon and energy footprints in the EU at 9.2tCO$_2$eq/cap and 0.14 TJ/cap per year (Figure 1). They contribute 17-18% of the EU's total carbon and energy footprints, but constitute less than 13% of the EU population. Two-person households are most numerous with 27% of the EU population. They also contribute the largest share of the EU's total carbon and energy footprints with 31-32%. The EU per capita average of carbon and energy footprints for two-person households amounts to 8.4 tCO$_2$eq/cap and 0.12 TJ/cap, respectively. The largest households (>4 persons) contribute about 9-10% to total EU emissions and energy use and represent 14% of the population. They have the lowest average carbon and energy footprints of 4.6 tCO$_2$eq/cap and 0.07 TJ/cap, respectively (Figure 1).

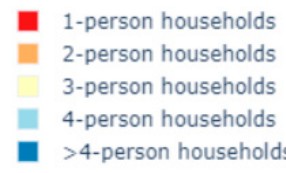

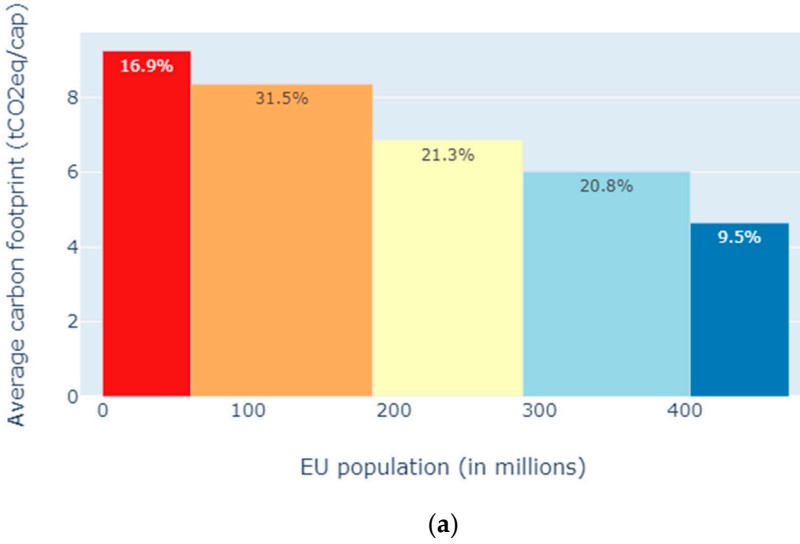

(**a**)

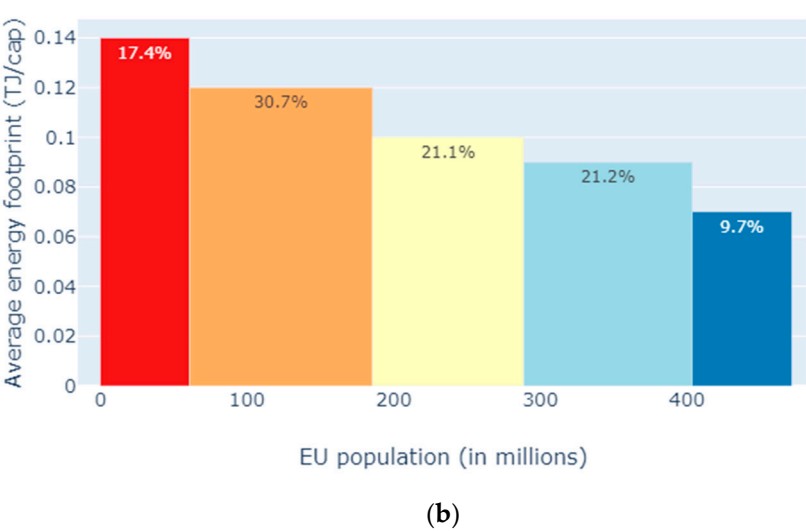

(**b**)

**Figure 1.** Distribution of EU carbon (**a**) and energy (**b**) footprint shares by household size. The total carbon and energy contribution can be split into two parts: the average carbon and energy footprints per capita (y-axis) and the number of people within the household cohort in the EU (x-axis). The %-s represent the share of total EU carbon and energy footprints by household sizes. Source: own calculations based on country population from the World Bank for 2010.

Figure 2 depicts the relationship between average per capita carbon and energy footprints and average household sizes across EU countries. The figure shows a negative trend across countries, suggesting a substantial overlap between countries with high average carbon and energy footprints and relatively low household sizes. The average household size in EU amounts to 2.4, varying between

2.2 and 2.9 across countries. The supplementary material (SM2) provides more detail about the distribution of carbon and energy footprints, and household sizes across EU countries.

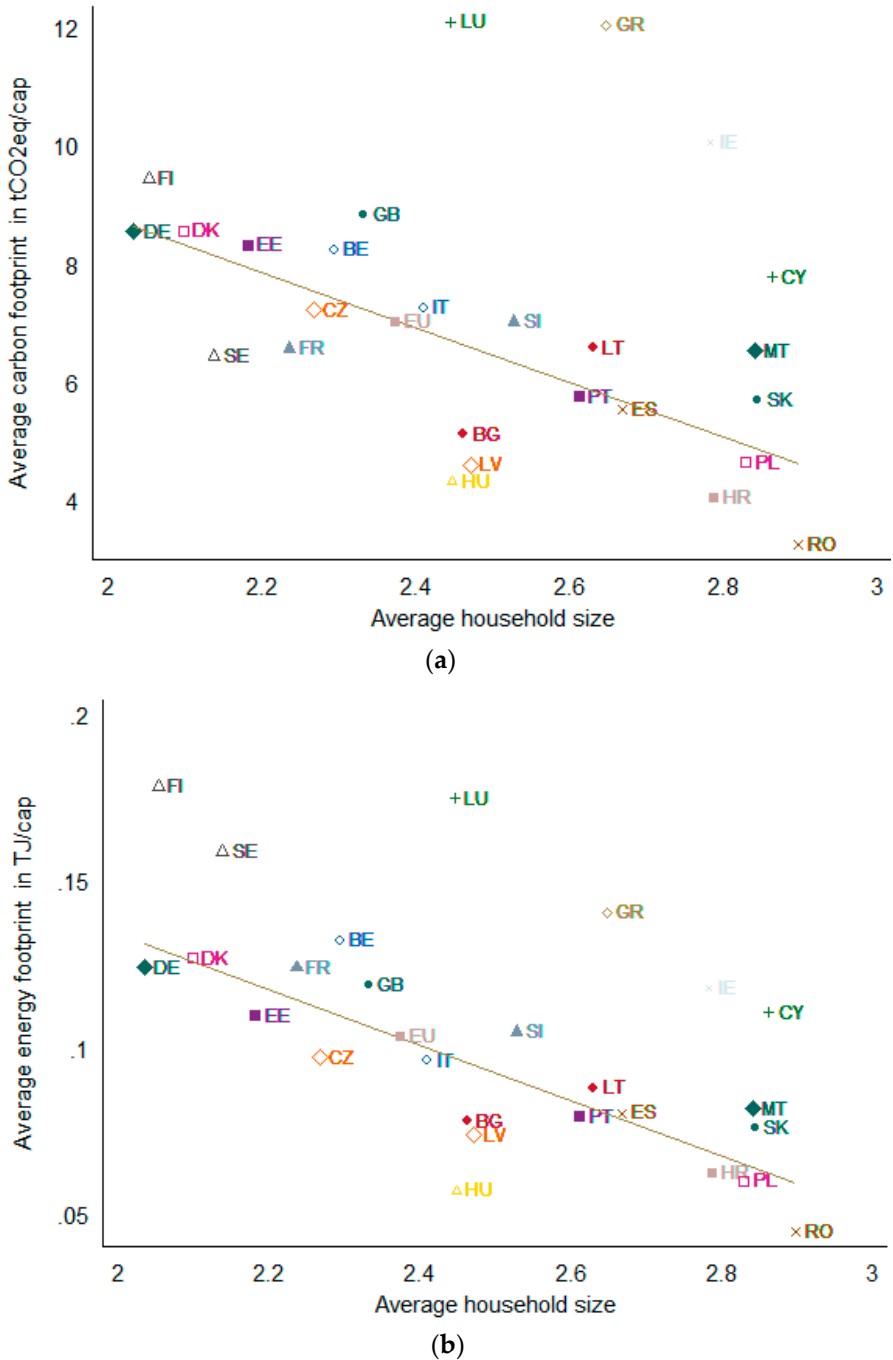

**Figure 2.** Association between average household size and average per capita carbon (**a**) and energy (**b**) footprints in the EU. The carbon footprints are measured in tCO$_2$eq/cap and energy footprints in TJ/cap. Household weights provided by the HBS have been applied.

The countries with the highest per capita carbon and energy footprints in the EU include Luxembourg, Greece (previously found to have one of the highest carbon footprints in the EU [4,43], with a large vessel fleet in relation to its size, requiring a high use of fuel from bunkers [45]), Ireland, Finland, United Kingdom, Belgium, Germany and Denmark, with carbon footprints between 14.1 and 9.1 tCO$_2$eq/cap, and energy footprints between 0.2 and 0.13 TJ/cap (Figure 2, SM2). These are also the

countries with some of the lowest household sizes: Germany (2.0), Denmark and Finland (2.1), Belgium and the United Kingdom (2.3). Finland and Denmark have the highest share of one-person households from the total number of households at 40%, followed by Germany at 39%. These observations broadly agree with the Eurostat statistics on household sizes (SM3).

The countries with the lowest per capita carbon and energy footprints include Romania, Croatia, Hungary, Latvia, Poland, Bulgaria, Spain, Portugal and Slovakia, with carbon footprints between 3.6 and 6.2 $tCO_2eq/cap$, and energy footprints between 0.05 and 0.09 TJ/cap. The countries with the highest household sizes include Romania and Cyprus (2.9), Slovakia, Malta, Poland and Croatia (2.8), and Spain (2.7). Romania, Malta and Spain have the lowest share of one-person households (19%) from the total number of households.

Figure 3 shows average per capita carbon and energy footprints per household size across EU countries. It confirms a drop in the environmental per capita impact with rising household size within EU countries. While the slopes vary in steepness, we consistently confirm this trend for all EU countries. For example, the average carbon footprint of Luxembourg ranges from 18.8 to 7.4 $tCO_2eq/cap$ for one-person and six-or-more persons households, respectively. Similarly, the per capita energy footprint of the average one-person household in Luxembourg is 0.27 TJ/cap, while that of an average six-or-more persons household amounts to 0.11 TJ/cap. According to Figure 3, the spread of the average carbon and energy footprints across EU countries is much larger for smaller household sizes compared to larger household sizes. Additionally, the absolute change in environmental impacts with the addition of one more household member is decreasing in magnitude with the rising household size.

3.1.2. Household Size and Population Density

The countries with lower average household sizes—Belgium, Germany, the United Kingdom and Finland—are also some of the most densely populated (Figure 4). At the same time, countries with larger average household sizes are more sparsely populated—e.g., Slovakia, Croatia and Poland.

Notable exceptions are Malta (with high average household size and a predominantly urban sample (92%) and Denmark (with low average household size and a largely rural sample, with as much as 43% of the sample living in sparsely populated areas). Denmark has a long tradition of a social-democratic welfare regime [46] with more liberal attitudes to family relationships and lower levels of religiosity, which may explain the relatively lower household sizes at lower population density. Compared to Western Europe, there is higher religious participation in Malta, attaching great importance to teachings regarding family life, the morality of abortion, divorce and other matters [47], which may explain the relatively large average household size. In addition, there may also be geographical reasons for the relatively high population density, with Malta being a small island.

Similar to other studies [23], we find that population density is important for per capita carbon and energy footprints (see SM2). Descriptive statistics should be interpreted with caution as they do not control for the differences in income levels and other relevant factors, which tend to vary substantially between urban and rural areas.

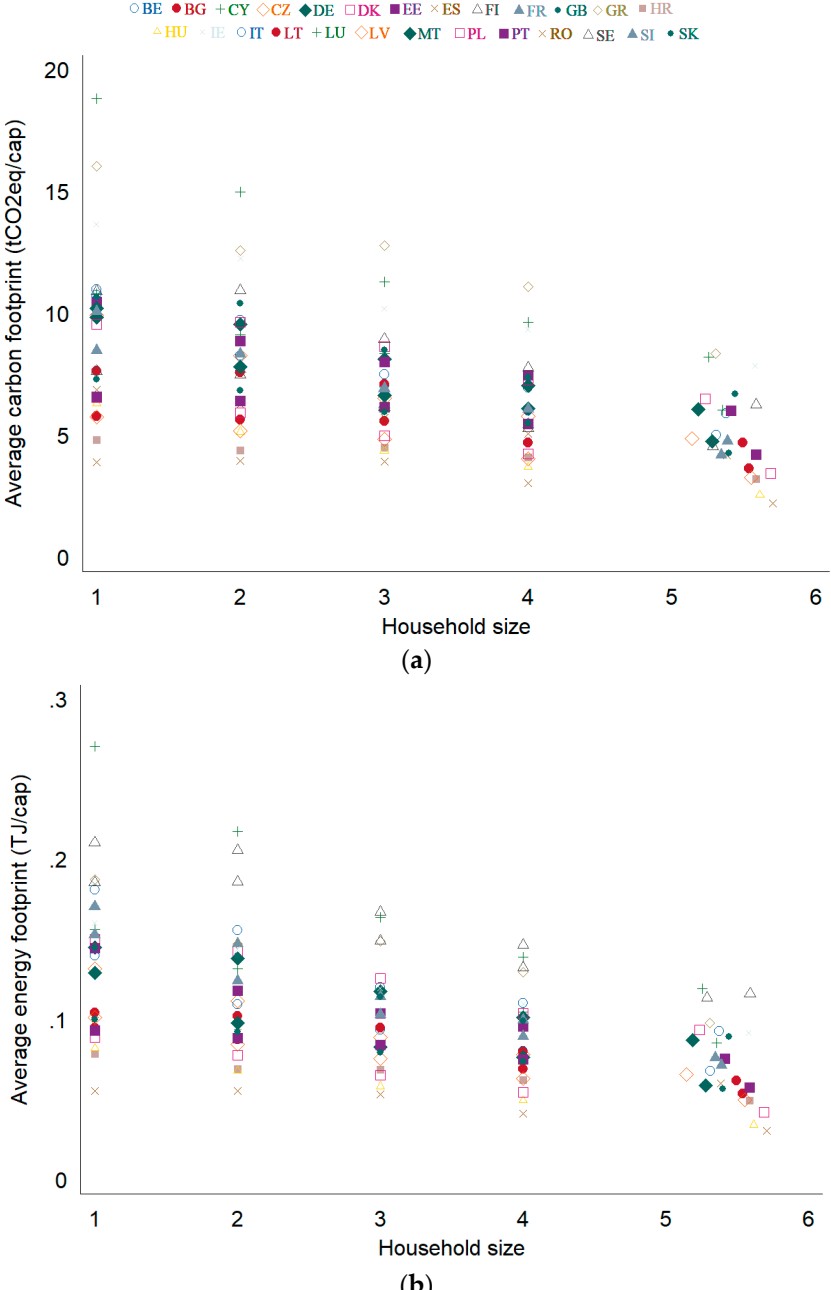

**Figure 3.** Mean per capita carbon (**a**) and energy (**b**) footprints by EU country by household size. Households with household sizes >5 have been aggregated in the same group. The carbon footprints are measured in $tCO_2eq/cap$ and energy footprints in TJ/cap. Household weights provided by the HBS have been applied.

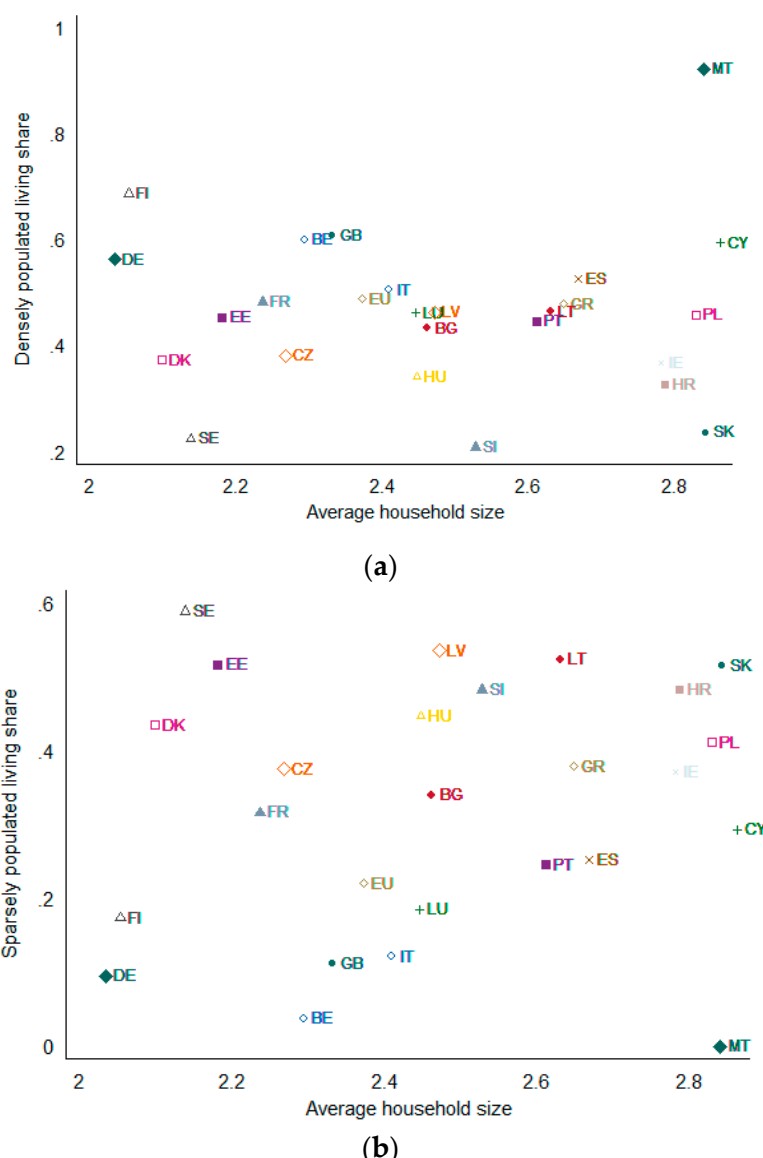

**Figure 4.** Association between average household size and share of households living in densely (**a**) and sparsely (**b**) populated areas in the EU. In cases where the shares do not add up to one, the difference amounts to the share of households living in intermediately populated areas. Household weights provided by the HBS have been applied.

### 3.1.3. Bi-Variate Regressions

Table 1 presents an overview of the standardized bi-variate Ordinary Least Squares (OLS) regression coefficients and statistical significance between household size (as a dependent variable) and urban-rural typology, carbon and energy footprints, and income per capita (as independent variables) across EU countries. Table 1 confirms a strong negative relationship between household size and per capita energy and carbon footprints within countries. The EU coefficients amount to −0.17 and −0.20 for carbon and energy footprints, respectively. Across countries, the coefficients vary between −0.11 (in Romania) and −0.39 (in Luxembourg) for carbon, and between −0.12 (in Romania) and −0.44 (in Czech Republic) for energy.

**Table 1.** Standardized bi-variate regression coefficients (can be interpreted as pairwise correlation coefficients) between household size and other variables by EU country.

| Country Code | Country Name | Coefficients for Household Size (HHSIZE) | | | | |
|---|---|---|---|---|---|---|
| | | Densely Populated | Sparsely Populated | Carbon Footprint | Energy Footprint | Income |
| EU | European Union | −0.040*** | 0.042*** | −0.170*** | −0.196*** | −0.208*** |
| BE | Belgium | −0.102*** | 0.011 | −0.310*** | −0.341*** | −0.297*** |
| BG | Bulgaria | −0.007 | −0.025 | −0.132*** | −0.195*** | −0.356*** |
| CY | Cyprus | −0.072*** | 0.01 | −0.274*** | −0.293*** | −0.276*** |
| CZ | Czech Republic | −0.083*** | 0.054** | −0.384*** | −0.437*** | −0.291*** |
| DE | Germany | −0.179*** | 0.114*** | −0.169*** | −0.182*** | −0.196*** |
| DK | Denmark | −0.114*** | 0.087*** | −0.162*** | −0.228*** | −0.154*** |
| EE | Estonia | −0.009 | −0.003 | −0.196*** | −0.241*** | −0.239*** |
| ES | Spain | −0.046*** | 0.016* | −0.191*** | −0.217*** | −0.416*** |
| FI | Finland | −0.150*** | 0.128*** | −0.159*** | −0.195*** | −0.181*** |
| FR | France | −0.082*** | 0.101*** | −0.229*** | −0.294*** | −0.242*** |
| GB | United Kingdom | 0.017 | −0.000 | −0.139*** | −0.141*** | −0.161*** |
| GR | Greece | −0.026 | −0.008 | −0.193*** | −0.190*** | −0.263*** |
| HR | Croatia | −0.095*** | 0.048** | −0.156*** | −0.200*** | −0.326*** |
| HU | Hungary | −0.147*** | 0.118*** | −0.282*** | −0.264*** | −0.415*** |
| IE | Ireland | −0.052*** | 0.082*** | −0.261*** | −0.236*** | −0.261*** |
| IT | Italy | −0.053*** | 0.013* | −0.273*** | −0.259*** | – |
| LT | Lithuania | −0.152*** | 0.150*** | −0.143*** | −0.146*** | −0.325*** |
| LU | Luxembourg | −0.109*** | 0.084*** | −0.391*** | −0.379*** | −0.391*** |
| LV | Latvia | −0.076*** | 0.076*** | −0.157*** | −0.224*** | −0.214*** |
| MT | Malta | −0.001 | – | −0.253*** | −0.245*** | −0.240*** |
| PL | Poland | −0.187*** | 0.144*** | −0.296*** | −0.306*** | −0.295*** |
| PT | Portugal | −0.02 | −0.044*** | −0.127*** | −0.134*** | −0.243*** |
| RO | Romania | – | – | −0.116*** | −0.122*** | −0.422*** |
| SE | Sweden | −0.011 | −0.002 | −0.184*** | −0.170*** | −0.231*** |
| SI | Slovenia | −0.079*** | 0.061*** | −0.277*** | −0.316*** | −0.179*** |
| SK | Slovakia | −0.098*** | 0.052*** | −0.202*** | −0.216*** | −0.355*** |

Note: * $p < 0.05$, ** $p < 0.01$, *** $p < 0.001$. The variables densely populated (*DENSE*) and sparsely populated (*SPARSE*) are dummies. In the context of this table, *HHSIZE* can be interpreted as a dependent variable, and the rest of the variables—as independent variables. Household weights provided by the HBS have been applied.

Furthermore, densely populated contexts (cities) are associated with smaller household sizes, and sparsely populated rural contexts – with larger household sizes in most EU countries (Table 1). The lowest regression coefficients between household size and densely populated context are found in Poland (−0.19) and Germany (−0.18). The opposite is true for sparsely populated areas, with the highest significant coefficient between household size and sparsely populated context found in Poland (0.14). Portugal shows an exceptional trend, being the only country with a negative and significant coefficient for household size and sparsely populated context.

While we note substantial inter-country differences, household dynamics should clearly be analyzed controlling for other socio-demographic trends (such as income and population density) [16]. For example, the analysis confirms a strong negative relationship between income and household size across all EU countries—suggesting an association between lower household sizes and higher incomes—with a coefficient amounting to −0.21 for the EU.

*3.2. Household Economies of Scale for Total Carbon and Energy Footprints*

Figure 5 portrays results from a multi-variate OLS regression on the role of *HHSIZE* for per capita carbon and energy footprints in logarithmic form (dependent variables). There are additional variables in the models such as income, urban-rural typology and geographical region (see the Data and Methods section for the model specification). The figure shows two model specifications, including (in blue) and excluding (in red) the *HHSIZE×DENSE* interaction terms.

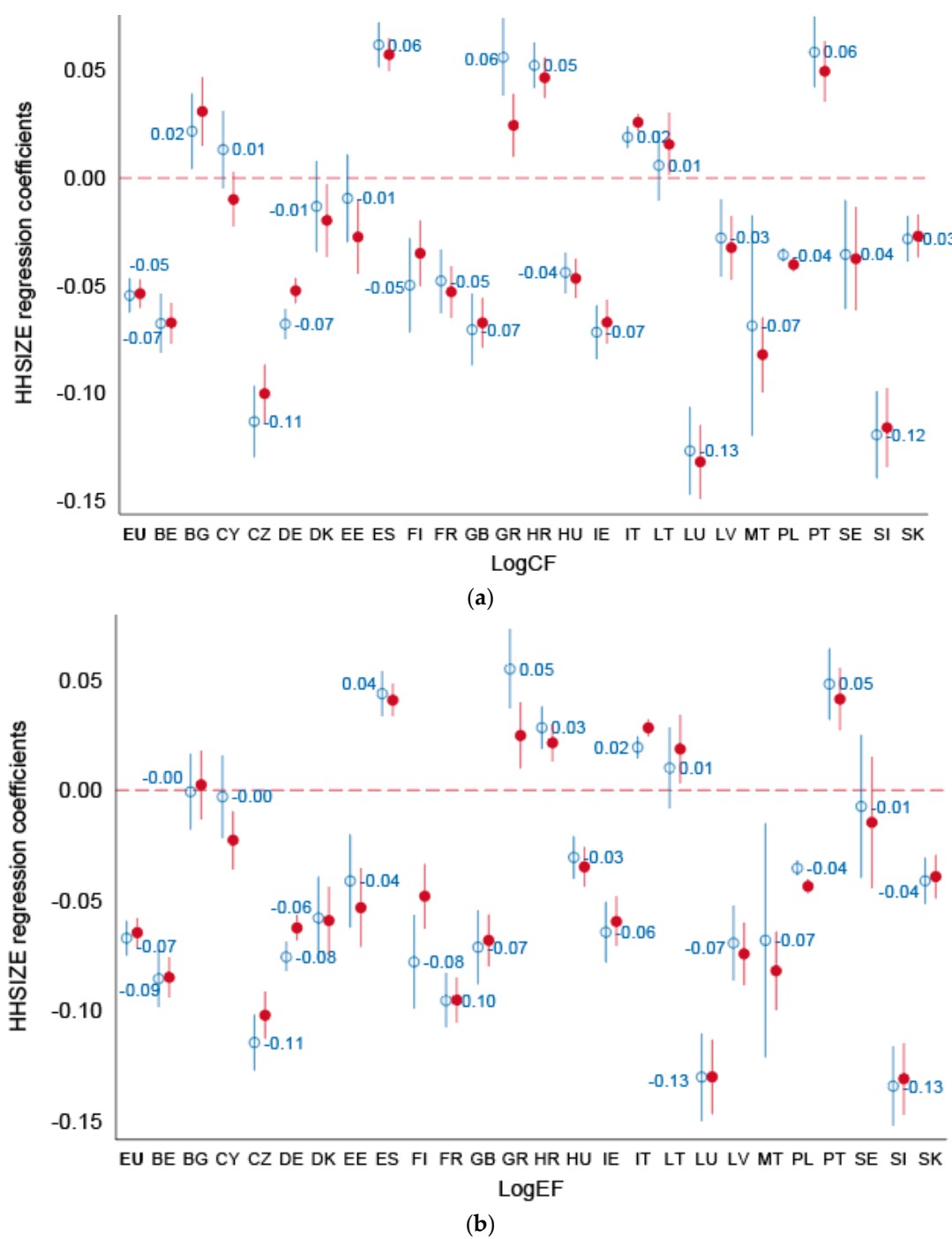

**Figure 5.** Household size effect across countries with dependent variables—the log of carbon footprints per capita (**a**) and energy footprint per capita (**b**). The blue coefficients depict the *HHSIZE* coefficient acquired from the model including interaction effect (*HHSIZE×DENSE*) and the red coefficient—the *HHSIZE* coefficient from the model without any interaction term. All models control for income, rural-urban typology and region. See Data and Methods for the model specification. Household weights provided by the HBS have been applied.

Figure 5 shows considerable variation across countries: while most countries display strong and moderate household economies of scale, there are also countries with no household economies of scale, or even with positive *HHSIZE* effects. Most countries (15 out of 25) in the EU sample show a negative and significant *HHSIZE* effect, which is in line with our initial hypothesis.

An increase in the EU household size by one member brings about a 5%-reduction in the carbon footprint and 7%-reduction in the energy footprint (Figure 5, in blue). The countries with the strongest household economies of scale include Luxembourg, Slovenia and Czech Republic, described by negative and significant *HHSIZE* at the 5% level coefficients, ranging from −0.11 to −0.13. The coefficients suggest that an increase in household size by one member decreases the per capita carbon and energy footprint by up to 12% (taking the exponent of the coefficient). Other countries—such as Belgium, Germany, Finland, France and the United Kingdom—are characterized by moderate household economies of scale. Their *HHSIZE* effects vary between −0.03 and −0.10, suggesting that an increase in the household size by one member reduces per capita carbon and energy footprints by 3–10%.

However, Figure 5 also points to countries—such as Cyprus and Lithuania—with no visible household economies of scale for the total carbon and energy footprint per capita. Against our initial hypothesis, several countries even stand out with positive and significant *HHSIZE* coefficients such as Spain, Italy, Greece, Portugal and Croatia. There are no significant differences between the *HHSIZE* coefficients for carbon and energy footprints in most countries (see SM Figure S9), suggesting similar economies of scale for energy and emissions.

The 95% confidence intervals of the *HHSIZE* coefficients in blue and red are also largely overlapping across EU countries, meaning that there is no significant difference of the *HHSIZE* effect magnitude regardless of whether or not the interaction term is included.

The following two sections explore these inter-country differences (1) for different consumption domains; and (2) in their interaction with population density. We consider contextual differences between countries to discuss these results in the Discussion section.

### 3.3. Household Economies of Scale by Consumption Categories

Figure 6 provides an overview of the *HHSIZE* regression coefficients across the various consumption categories with the logarithm of the carbon footprint by consumption category as a dependent variable. We note substantial differences between EU countries within each consumption category, both in terms of household economies of scale and carbon contribution. Figure 7 shows the *HHSIZE* regression coefficients and their 95% confidence intervals across the EU countries. A detailed overview of the sectors included in each consumption category can be found in the supplementary material.

The coefficient ranges highlight the differences of the magnitude of household economies of scale and point to some of the products and services associated with higher sharing rates compared to others. For example, the strongest household economies of scale are noted for housing categories such as rents and mortgages, electricity and household services. These housing categories have median carbon shares of 5%, 8% and 4%, respectively (Figure 6). At the same time, some of the weakest household economies of scale are noted in the transport domain, which is also characterized with the highest median carbon share of 25%.

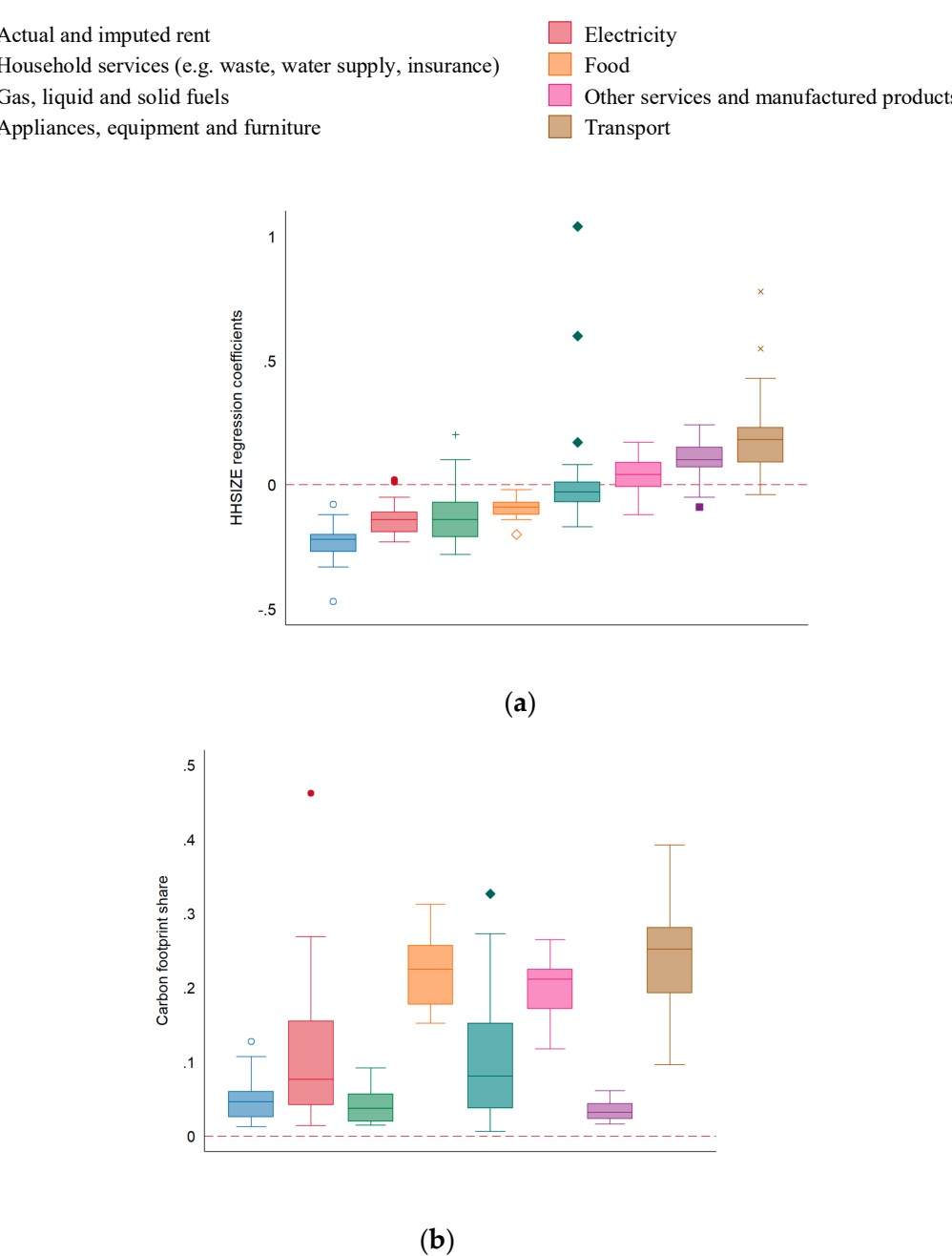

**Figure 6.** A summary of the *HHSIZE* regression coefficients of EU countries (displayed on Figure 7) with the logarithm of the per capita carbon footprint by consumption category as dependent variables (**a**) and the proportion of the individual consumption categories of the overall carbon footprint of EU countries (**b**). The categories are ordered by the median *HHSIZE* effect depicting the importance of the household economies of scale from the strongest to the weakest.

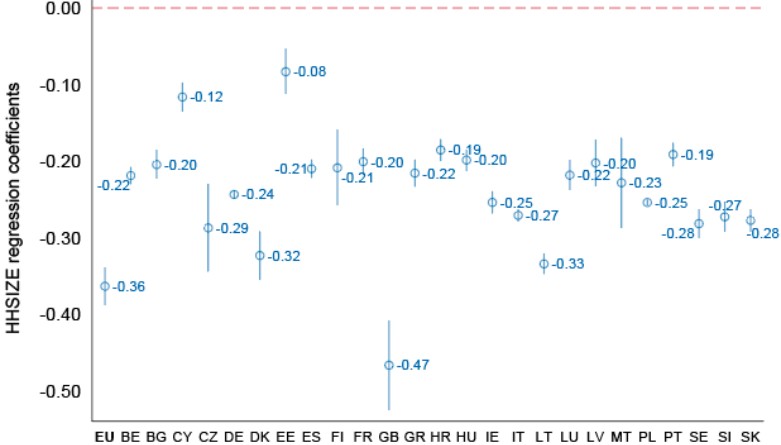

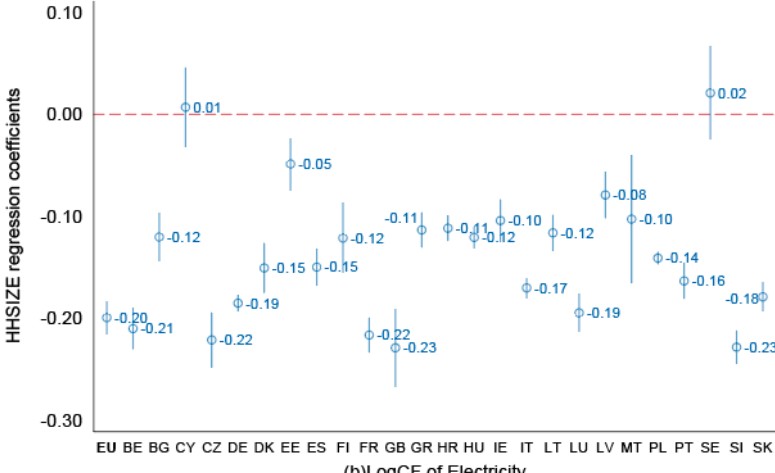

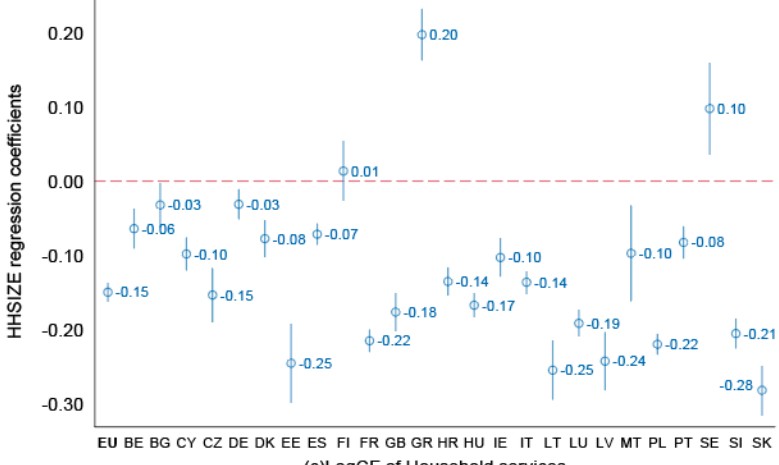

**Figure 7.** *Cont.*

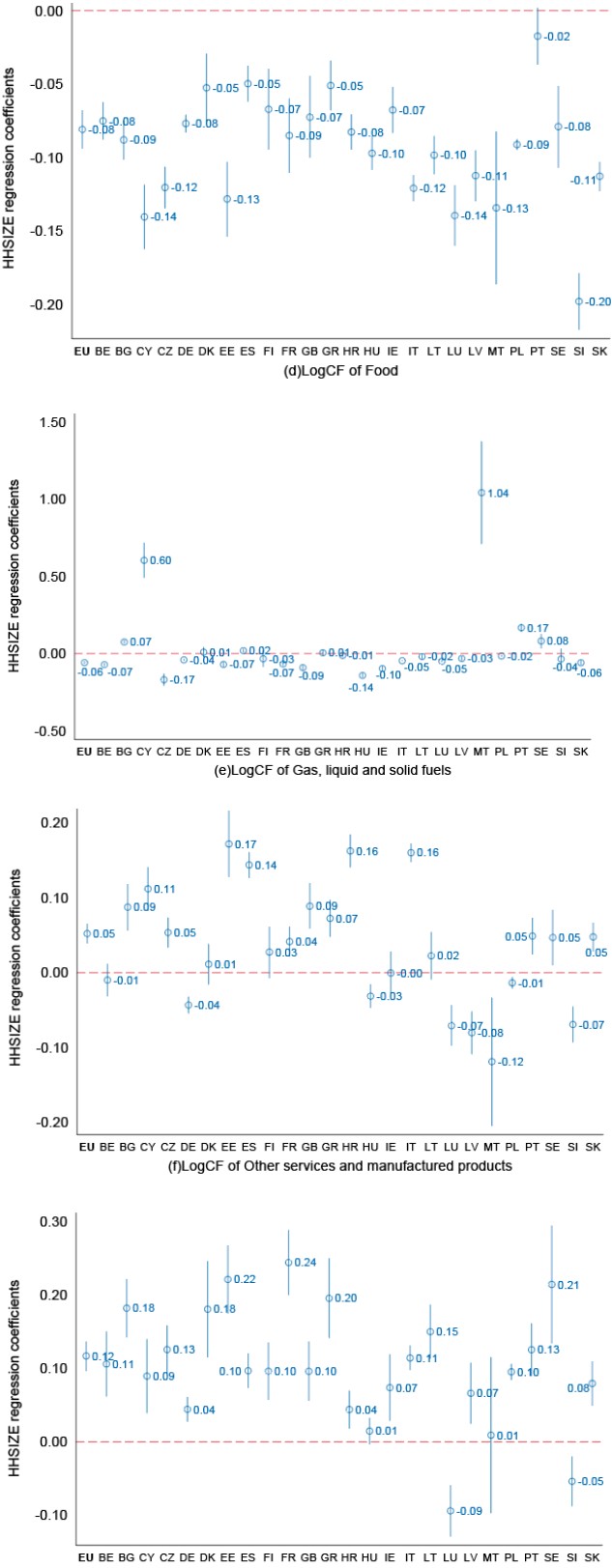

**Figure 7.** *Cont.*

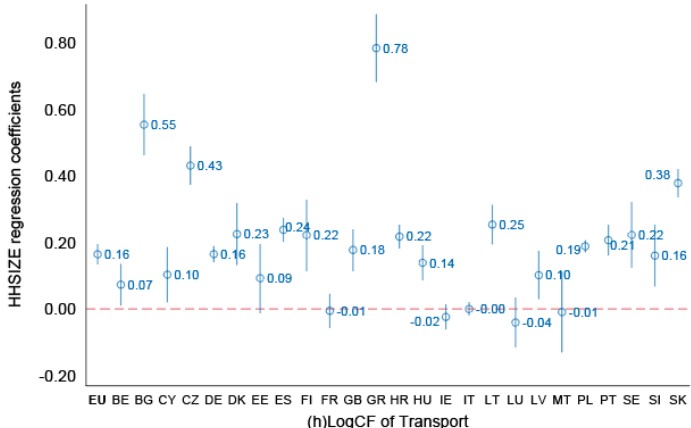

**Figure 7.** Regression coefficients for household size effects on the logarithm of the per capita annual carbon footprint by consumption category. Categories: (**a**) Actual and imputed rent; (**b**) Electricity; (**c**) Household services, e.g., waste treatment, water supply, insurance; (**d**) Food; (**e**) Gas, liquid and solid fuels; (**f**) Other services and manufactured products; (**g**) Appliances, equipment and furniture; and (**h**) Transport. The categories are ordered by the median *HHSIZE* effect depicting the importance of the household economies of scale from the strongest to the weakest. Household weights provided by the HBS have been applied.

### 3.3.1. Housing

Substantial household economies of scale are noted for home- and housing-related categories, particularly housing rent or real estate services (Figure 7(a)), electricity (Figure 7(b)) and household services such as waste treatment, water supply and insurance (Figure 7(c)). The *HHSIZE* effect associated with rents and mortgages vary between −0.08 (for Estonia) and −0.47 (for the United Kingdom) (the category includes development of building projects, management and support services). This means that an increase in the household size by one member is associated with an 8–37% reduction (taking the exponent of the coefficient) in the carbon footprint associated with real estate services. With regards to electricity, negative and significant *HHSIZE* coefficients between −0.05 for Estonia and −0.23 for the United Kingdom and Slovenia are noted; this suggests a 3-21% reduction in the related per capita carbon footprint with an additional household member. Cyprus and Sweden stand out with insignificant *HHSIZE* effects (the Swedish HBSs offered a lower level of consumption detail aggregating all home-related energy consumption). Similarly, strong household economies of scale are noted in terms of household services with the largest (negative) coefficients noted for Slovakia (−0.28), Lithuania and Estonia (−0.25). That is, the increase of household size by one member results in a reduction of the household services emissions by as much as 24%.

While similar ranges of the household economies of scale are noted for electricity and housing fuels (Figure 6), the strong positive outliers in terms of *HHSIZE* effects lower the median household economies of scale for housing fuels. We found negative and significant *HHSIZE* coefficients varying between −0.17 (for Czech Republic) and −0.04 (for Germany and Slovenia) across most EU countries (Figure 7(e)). The positive and significant effects—especially for Malta and Cyprus—could potentially be explained by product allocation inconsistencies of fuel use from marine bunkers [45] (where we do not expect household economies of scale) being inaccurately allocated to household fuels in the national accounts.

The strong household economies of scale in the household domain are in line with prior claims that household size is one of the largest determinants of domestic energy consumption [48] and shelter carbon footprints [4]. They result from the sharing of space and embodied energy in buildings, energy for heating, cooling, lighting and shared appliances and activities [9].

### 3.3.2. Food

Food-related economies of scale in larger households may occur when household members prepare (e.g., when sharing food ingredients) and manage food together (e.g., when they better manage food waste [49], which we were not able to test in this study). Furthermore, larger households may be more likely to buy food in larger quantities, which may cost less per unit [50]. While this may allow for a reduction in embodied emissions, e.g., through reduced packaging, in our model we were unable to capture any differences in carbon intensities within food products. As we applied monetary-based carbon intensities, any reduction in food spending due to lower price is reflected in our model in lower carbon footprints, which may be misleading in cases of large price variation within products. Finally, there may be other carbon reduction potential associated with the use of common utensils, appliances for cooking and storing food and shared shopping for larger households. These effects are included in the estimates for housing and transport in our analysis.

Figure 7(d) denotes significant negative coefficients between −0.20 (for Slovenia) and −0.05 (for Denmark, Spain and Greece), suggesting that an increase in the household size with one member leads to a decrease in the food-related carbon footprint by 5–18%.

### 3.3.3. Equipment, Transport and Other Consumption

While we expected substantial household economies of scale for shared appliances, equipment and furniture, we find that most EU countries report positive *HHSIZE* regression coefficients (Figure 7(g)). A potential explanation of this result is that while some appliances, machinery and furniture are shared within households, the sectoral detail of EXIOBASE does not allow us to distinguish between typically shared and individually-used items. Furthermore, this category only includes the purchase of items (and hence their embodied carbon footprint), while the direct emissions associated with the use phase is included in the analysis of electricity and housing fuels. Notable exceptions with moderate household economies of scale for home appliances and equipment include Luxembourg and Slovenia with regression coefficients of −0.09 and −0.05, respectively.

We did not find consistent household economies of scale for transport—with positive or insignificant coefficients for all EU countries (Figure 7(h)). Larger households have potential to stabilize car ownership [51,52], where additional household members do not require additional number of cars. Prior longitudinal analysis of French car sharing practices shows that while household car sharing is a regular practice concerning almost half of the French car fleet, this trend is decreasing [53]. Their analysis further highlighted gender differences in terms of car sharing within households, with a higher proportion of main users being male and a higher proportion of secondary users being female [53].

However, our analysis suggests that the benefits of shared travel within the household are not realized in many countries in Europe (Figure 7(h)). The lack of household economies of scale with regards to personal vehicles and equipment (SM4) suggests that additional household members may also activate a need for another household car, e.g., following a partnership formation [54]. There may also be offsetting effects such as using the car more intensively or having a larger car in single-car households [55]. Furthermore, no household economies of scale were noted for other transport modes such as air travel, for which there is a growing demand with rising incomes in Europe [31].

Finally, we did not observe substantial household economies of scale with regards to other services and manufactured products (Figure 7(f)). There may also be additional factors that strongly correlate with household size (e.g., demographic, social, cultural and economic characteristics) that we could not include in our model due to lacking data, which may explain the variation in coefficients.

### 3.4. Household Size and Population Density Interaction

In this section, we discuss the magnitude and significance of interaction effects depicted in Figure 8 (*HHSIZE×DENSE)* across EU countries (the model in blue in Figure 5, controlling for income,

household size, population density and region). The majority of EU countries show insignificant interaction coefficients, suggesting no significant differences in the *HHSIZE* effect between densely and sparsely populated areas in the EU.

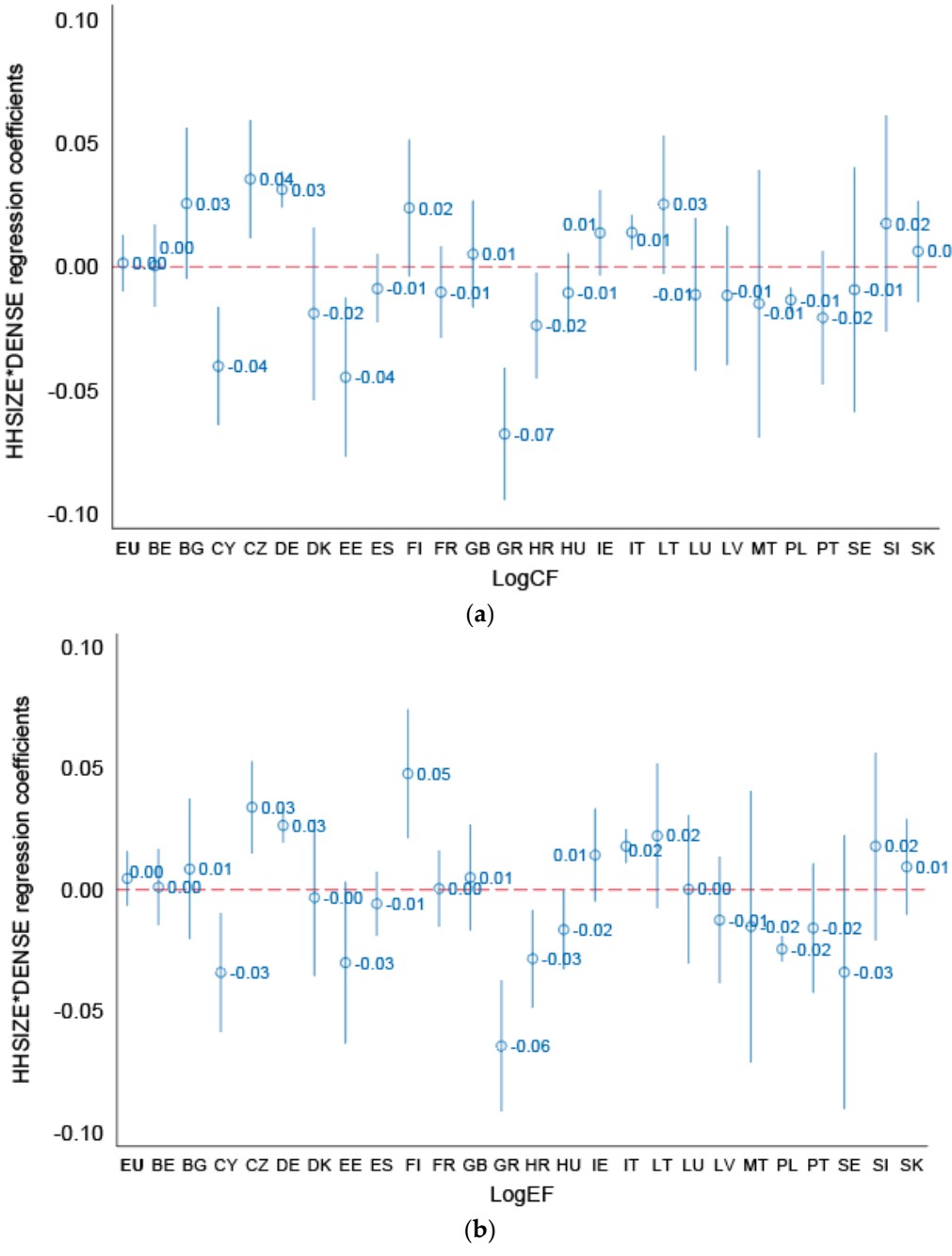

**Figure 8.** Interaction effect between household size and population density (*HHSIZE×DENSE*) across countries. Dependent variables—the log of per capita carbon (**a**) and energy (**b**) footprints. We excluded the *HHSIZE×INTERMEDIATE* interaction term as no significant differences of the household size effects between intermediately and sparsely populated areas were noted. See Data and Methods for the model specification. Household weights provided by the HBS have been applied.

However, several countries such as Czech Republic and Germany demonstrate negative *HHSIZE* effects and positive interaction effects (*HHSIZE×DENSE*), both of which are significant at the 5% level. This result suggests that adding another household member in a sparsely populated (rural) environment is associated with larger household economies of scale compared to doing so in a densely

populated (urban) environment. While adding a household member to a rural household reduces per capita energy footprints by 7–11%, adding a household member to a dense urban household reduces them by 4–7% (Figure 5, Figure 8). The lower household economies of scales in densely populated environments are noted particularly for electricity, housing fuels, appliances, equipment and furniture, and food. This is plausible because these types of environmental impacts tend to be higher in rural areas in these countries, compared to urban areas, so that greater household sizes can reduce these impacts more in rural areas.

We find negative interaction effects for other countries, particularly for Greece, Estonia, Cyprus and Croatia, suggesting that households in densely populated regions encounter higher household economies of scale in these countries, compared to sparsely populated areas. The analysis of consumption categories suggests that these negative interaction effects are primarily associated with consumption of household services (e.g. water and waste), other services and manufactured products.

These negative interaction effects may contradict our previous hypothesis that the interaction between household and urban economies of scale leads to higher household economies of scale in rural and sparsely populated areas [24,25]. However, these are all countries where per capita environmental footprints tend to be higher in urban compared to rural areas (SM2), so it is plausible that adding a household member in urban areas leads to greater reductions of per capita environmental footprints there compared to rural areas.

## 4. Discussion and Conclusions

### 4.1. Household Dynamics within the EU

One-person households are the most carbon and energy intensive in per capita terms, contributing to 17-18% of the EU total carbon and energy footprint. The per capita carbon and energy footprint of a one-member household is about twice that of a five- or more person household in the EU. The share of those living in one-person households varies from 40% in Finland and Denmark to 19% in Spain, Malta and Romania, with an EU average of 31% from the total number of households.

We note substantial differences in household sizes across various EU countries as well as the role of household size for per capita carbon and energy impacts. Adding an additional household member results in a carbon and energy reduction of above 10% on average in some EU countries. This result confirms that shrinking household sizes across the EU and globally are of key concern for climate change mitigation. They should thus be adequately considered in modelling work, e.g., prospective scenarios of socio-demographic trends and their influence on carbon and energy footprints and pathways to meet carbon targets. Household dynamics should also be regarded in the context of mitigation solutions and experimentation with alternative household formations.

Substantial differences in the household economies of scale are noted for various consumption domains, with a higher potential in housing-related items such as electricity use (up to 21% reduction with an additional household member), real estate services (up to 37%) and household services such as waste collection and water supply (up to 24%). Food and fuel consumption show moderate household economies of scale with up to 18% reduction of the carbon footprint with an additional household member in some EU countries. We note lower or no household economies of scale in other domains of consumption (e.g., transport, manufactured products and services), where an increase in the household size likely corresponds to an increase in consumption needs (e.g., second household vehicle, more clothing, educational or health services with an additional household member).

Furthermore, the majority of EU countries have comparable household economies of scale between urban and rural areas (insignificant interaction term). Other countries such as Czech Republic and Germany report higher household economies of scale in sparsely populated areas, in line with prior evidence [24] (positive interaction term). We also found a negative interaction effect between household size and population density for a third group of countries, which counters our original hypothesis; yet,

these are countries in which per capita emissions and energy use tend to be higher in urban compared to rural areas, unlike most other EU countries.

## 4.2. Country Clusters and Contextual Factors

Table 2 summarizes our observations regarding the household economies of scale by various consumption domains and the interaction with population density. Two clusters of countries emerge—one with strong or moderate household economies of scale, and one with lower or no household economies of scale.

**Table 2.** A summary of country clusters with regards to household economies of scale and other contextual differences.

| | Country Clusters | Example Countries | Mean Household Size and T-test | Household Economies of Scale by Consumption Domains | Interaction with Population Density |
|---|---|---|---|---|---|
| 1 | Countries with high/moderate/low household economies of scale | LU, SI, CZ, BE, DE, FI, FR, GB, MT, DK, HU, IE, LV, PL, SE, SK | 2.54 (0.003) | Strong household economies of scale for actual and imputed rent (GB, CZ, DK, SE), electricity (GB, BE, CZ, DK, FR, SI), household services (SK, LV), food (MT, SI, LU), housing fuels (CZ, HU), other goods and services (MT, LU, LV), appliances and equipment (LU, SI); | Higher household economies of scale in rural areas compared to urban areas (DE, CZ) |
| 2 | Countries with no household economies of scale/Countries with positive *HHSIZE* effect | CY, LT, EE, ES, IT, GR, PT, HR, BG | 2.64 (0.005) | Some of the lowest household economies of scale (or positive coefficients) for actual and imputed rent (EE, CY), electricity (CY), household services (GR), food (PT, ES, GR), housing fuels (CY), other goods and services (EE, ES, LT, IT), appliances and equipment (EE, BG, GR, LT, IT) and transport (GR, BG); | Higher household economies of scale in urban areas (GR, EE, CY, HR), relatively low share of urban population and higher environmental impacts in urban areas. |
| | Difference | | *** | | |

Note: One-sided two-sample unweighted t-test is performed in order to compare the average household sizes between the country clusters under the following hypotheses: $H_0$: $\mu_{cluster2} - \mu_{cluster1} = 0$, $H_A$: $\mu_{cluster2} - \mu_{cluster1} > 0$. We estimated separate variances to control for significant differences in sample sizes between the country clusters. Standard errors are presented in parenthesis. T-test significance levels: * $p < 0.1$, ** $p < 0.05$, *** $p < 0.01$.

The first cluster—with high and moderate household economies of scale—consists of predominantly Northern and Central European countries. An increase in the household size by one member results in a reduction of the total carbon and energy footprint by 3–13% (Figure 5). This cluster is characterized by strong welfare regimes that promote individual independence and female labor market participation [19]—which may explain the lower household sizes in these countries. The cluster includes Belgium, Denmark, Sweden, Finland, France, Germany and the United Kingdom, which are similar in terms of socio-demographic context [56]. The small countries of Malta and Luxembourg are exceptions in terms of welfare regime [47,56]; the regression coefficients of Malta in particular are characterized by relatively high error ranges across most consumption categories, and results should thus be interpreted with caution. Finally, the Czech Republic, Poland, Slovakia, Slovenia and Hungary (and Croatia, which is allocated to the second cluster in terms of household economies of scale in our analysis) are characterized by the Central Europe welfare model, associated with lower income inequality, lower rates of unemployment, higher labor market flexibility and higher social contributions and government expenditure as a share of the Gross Domestic Product compared to the Eastern European countries in the second cluster [56].

The second cluster—with lower or no household economies of scale—consists of predominantly Southern and historically Catholic countries as well as some Eastern European states. An increase in the household size by one member does not change the total per capita carbon and energy footprint, or even increases in the per capita environmental impact (Figure 5). These countries already have higher household sizes, and emphasize the role of the family for mutual support or are more "collectivistic". Greece, Spain, Italy, Cyprus and Portugal stand out from other EU countries in terms of their welfare regimes previously described as the Mediterranean welfare model [56] with stronger influence of Catholicism and traditional family values [57–59]. The Eastern European welfare model—including Lithuania, Estonia and Bulgaria (and also Latvia, which is included in the first

cluster in our analysis)—is associated with strong nuclear family institutions, low social protection expenditure primarily on old-age pensions, high income inequality, rigid and discriminatory labor markets and lower government capacity for generous social policies [56,60]. This might also contribute to higher family dependency for financial and welfare support, and hence higher household sizes. The reliance on extended family for assurance against risks of ill health, unemployment or poverty could be reduced with higher standards of living and the provision of stronger social-security systems in these countries [61].

These clusters show significant differences in terms of the average household sizes, with the second cluster denoting a significantly higher household size (Table 2, Figure 2). Considering the decreasing rate of household economies of scale with rising household sizes, this may partly explain the lower household economies of scale in these countries—where there already is a lot of within-household sharing, thus, there is less to gain by adding a household member.

Heating degree days are positively correlated with the housing-related energy use (and carbon footprints), with dwellings in colder regions requiring more energy to heat over the year [4]. This effect is partly mediated by stricter building standards in northern European countries, which reduces the amount of energy for heating per heating degree day [62]. Nevertheless, colder countries are likely to report higher household economies of scale particularly due to the high importance of home energy for the overall household economies of scale, which is also in line with the country clustering. This might also explain why we find significant positive *HHSIZE* effects in countries such as Spain, Italy, Greece, Portugal and Croatia. Not only are these countries with relatively large average household sizes already (and hence less scope for further within-household sharing), but there is also less of a requirement for heating, which is associated with some of the strongest capacity for household economies of scale.

The positive *HHSIZE* coefficients for some of the categories, where we expect relatively low possibilities for sharing is likely driven by other socio-demographic, infrastructural and economic factors that vary with household size, that we cannot explicitly control for in our model because they are not captured in the HBSs.

### 4.3. Policy Recommendations

Targeting the trend towards smaller households and under-occupation of homes in the EU and globally is a key option to reduce per capita carbon and energy contributions, with a higher mitigation potential compared to efficiency improvements such as upgrading the thermal insulation or more efficient appliances [42,63]. Understanding needs and expectations about personal space as well as changing social norms [18] are key for the upscaling of "downsizer homes" [63] and other alternatives to encourage within household sharing. Household sharing has an important gender dimension [53] as well; sharing may support the depersonalization of objects allowing for them to be managed and used jointly, thus encouraging even more (and more gender equal) sharing [53].

Yet, the trend of smaller households results from a myriad of processes, some of which cannot be reversed (e.g., falling birth rates or liberation from norms), or which we consider valuable for other reasons (e.g., female emancipation, financial independence or residential autonomy) [48,61]. For example, higher divorce rates worldwide may result in an increase of energy use and GHG emissions per capita [13]; however, the freedom to divorce is also a matter of human rights and social justice. This makes it crucial for policy interventions to realize the complexity of household dynamics and the inter-connections with social and environmental wellbeing.

Proximate causes of the reduction in household sizes worldwide include lower fertility rates, higher divorce rates and a decline in the frequency of multi-generational families with increasing non-family provision of care among others [61,64]. There is some evidence that the trend of decreasing fertility rates and increasing divorce rates is reversing since the early 2000s [65,66], which may also stabilize or even reverse the trend of smaller households. This suggests that the trend towards smaller families over the past half century did not result from a lasting change of family preferences, but rather

from a change in women's roles and labor market participation when institutions and partnerships had not yet adapted [65]. To successfully promote parenthood and female labor force participation, there is a need for a strong investment in childcare services, flexible workplace support and other family support [65,66]. Such policies may help reconcile work and family responsibilities and promote gender equity [65].

Additional social and psychological factors that may have influenced the reduction in household sizes include liberation from strict norms, less religiosity and increased importance of individual autonomy, self-actualization and privacy [48,61]. Support and increased visibility [67] for alternative household types—such as intentional communal living—may encourage larger households, which share lifestyles, cultural elements and common sense of purpose. Such alternative forms of living may thus be less challenging in terms of these social and psychological factors [12], compared to traditional family living. Yet alternative living arrangements may also be associated with difficulties in negotiating common and personal items, space and time [9]. Partnerships between policymakers and sharing initiatives may help tackle such difficulties by alleviating structural and institutional constraints and reducing social distance (e.g., by fostering care for the community) and geographical distance (e.g., by improving connectivity), which impede sharing [9,11]. Sharing emerges as an opportunity to act collectively on growing social, political and environmental awareness and steadily transforms social norms and routines [68].

The complexity of household dynamics and the low household economies of scale in high-carbon consumption domains such as transport encourage the consideration of additional ways to share resources between households as well. For example, while sharing a car may reduce the energy use and emissions associated with travel *within* the household (particularly in car-dependent areas outside urban cores [22]), in the presence of an excellent public transport system, the mitigation potential may actually be higher through sharing *between* households. Further research on household sharing in the context of public infrastructure and sharing initiatives at a higher spatial resolution—in both urban and rural context—is needed to explore the carbon mitigation potentials associated with sharing. Such wider sharing practices for de-carbonization and low energy demand require the provision of social and technological infrastructure such as investment in public spaces, green areas, mass transportation and new forms of peer-to-peer sharing [9,24]. The establishment of collective systems (e.g., universal basic services [69])—as opposed to highly individualized energy service delivery—also enables more resilient societies and prevents future emission lock-in [70].

In this paper, we explore possible impacts of household dynamics on per capita emissions, and examine difference in within household economies of scale across EU countries. Our main finding is that household economies of scale vary substantially across consumption categories, urban and rural typology and EU countries. We identify potential explanations associated with the sharing potential of various products and services, contextual differences in terms of social and cultural norms, geographic context, infrastructural and political context. Targeting trends towards smaller households and under-occupation of homes and encouraging sharing offers substantial potential to mitigate climate change with already available technologies and infrastructure.

**Supplementary Materials:** The following are available online at http://www.mdpi.com/1996-1073/13/8/1909/s1, SM1: Household Budget Surveys, SM2: Descriptive statistics by countries, SM3: Eurostat statistics, SM4: Total carbon and energy footprint determinants, Supplementary spreadsheet including the overview of consumption categories.

**Author Contributions:** Conceptualization, D.I. and M.B.; Formal analysis, D.I.; Funding acquisition, D.I. and M.B.; Investigation, D.I. and M.B.; Methodology, D.I. and M.B.; Software, D.I.; Supervision, M.B.; Visualization, D.I.; Writing—original draft, D.I.; Writing—review and editing, M.B. All authors have read and agreed to the published version of the manuscript.

**Funding:** This research was funded by the European Union's Horizon 2020 research and innovation program under Marie Sklodowska-Curie grant agreement, grant number 840454. The authors also received support from the UK Research Councils under the Centre for Research on Energy Demand Solutions.

**Acknowledgments:** We thank the whole EXIOBASE team for the effort to build the database and make it available for other researchers to use. In particular, we would like to thank Richard Wood for his assistance in the early stages of the environmental footprint analysis and Arkaitz Usubiaga-Liaño for his effort in compiling and communication the energy extensions. We would also like to thank Sylke Schnepf and two anonymous reviewers for their valuable feedback.

**Conflicts of Interest:** The authors declare no conflict of interest.

**Data Statement:** The data associated with this paper is available from University of Leeds at https://doi.org/10.5518/785. The dataset includes the per capita carbon and energy footprint calculations (generated by the authors of this study) together with household and country IDs from the HBS dataset disseminated by Eurostat. Please use the following data citation when referring to the dataset [71]: Diana Ivanova and Milena Büchs (2020): Carbon and energy footprints of European households (EU HBS) University of Leeds. [Dataset]. https://doi.org/10.5518/785.

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
