# Peer review of "Household Sharing for Carbon and Energy Reductions: The Case of EU Countries"

_energies, doi:10.3390/en13081909_

Round 1

Reviewer 1 Report

This is a comprehensive and interesting comparative study on the interactions of household size and the climate related environmental effects of household consumption. The spatial coverage of all contemporary member states of the EU clearly fills a comparative gap in the field of environmentally extended household consumption studies. The authors have well related their results to broader social trends and cultural contexts. The text is well written and easy to follow, and I enjoyed reading the manuscript. Despite of a strong basis, there is one major and some minor issues to consider.

The main consideration is related to the applied spatial scales and the resulting claims. The authors should seriously consider and conceptualise the spatial unit of exploring the economies of scale in different spatial contexts. Their current approach linking the average density of an undefined area with the potential of sharing between households (i.e. on close community and neighbourhood level) is simplistic and problematic due to the serious mismatch of spatial scales. Although the authors claim that the methodology of household consumption survey is described elsewhere, they should clearly indicate how the density variable has been originally collected or assigned, and which spatial unit it represents (this is not evident from their given Eurostat HBS reference either). E.g. municipality average density is a very different proxy from the scale of neighbourhood density or even building type where the sharing between households may occur. In addition, the authors ignore settlement size, hierarchies and the related economies of scale that shape the contexts of opportunities to share (incl. the presence of various shared mobility schemes). This is related to the core distinction “urban-rural” of the dataset itself, which is a strong simplification, missing the complex and interlinked functionality, and population distribution of regions. If the authors miss relevant spatial data, they should be very cautious with their related claims. PS Section 3.3. and the supplementary material could depict the differences between consumption categories across spatial scales.

See further on the choice of spatial scales or understanding shared mobilities:

Burgalassi, D, Luzatti, T. 2015. Urban spatial structure and environmental emissions: A survey of the literature and some empirical evidence for Italian NUTS 3 regions. Cities 49: 134–148, https://doi.org/10.1016/j.cities.2015.07.008

Freundendal-Pedersen, M., Kesselring, S. 2018. Sharing mobilities. Some propaedeutic considerations. Applied Mobilities 3: 1–7. https://doi.org/10.1080/23800127.2018.1438235

Jones, C., Kammen, D.M. 2013. Spatial distribution of U.S. household carbon footprints reveals suburbanization undermines greenhouse gas benefits of urban population density. Environmental Science & Technology 48: 895–902. https://doi.org/10.1021/es4034364

Poom, A., Ahas, R. 2016. How Does the Environmental Load of Household Consumption Depend on Residential Location? Sustainability 8: 799. https://doi:10.3390/su8090799

Household type and structure. The paper would benefit from a more structured approach to household type and structure. As consumption demand and pattern is age group specific, the authors could provide some discussion (if it is out of the scope of the analysis) on how the household structure affects the economies of scale. E.g. does the effect of adding another household member differ when (s)he is from an older, third generation or a newborn baby? Could there be a difference in one- and two-parent families?

Transportation. The results are very interesting and as they work against normal understanding of the benefits of shared commodities, the reasoning should be stronger. E.g., could the positive association between household size and footprint lean among others on the phenomenon that an additional household member induces a need for another household car with respective costs.

See also:

Ritter, N., Vance, C. 2013. Do fewer people mean fewer cars? Population decline and car ownership in Germany. Transportation Research Part A: Policy and Practice 50: 74–85. https://doi.org/10.1016/j.tra.2013.01.035

Housing. In the contemporary world (and Europe), people are more mobile either on personal or work-related reasons and they might have several homes, including in transnational situations. On the other hand, in several European countries, there is a tradition of having second homes (summer cottage, childhood home without permanent residents). How do the HBS dataset and the analysis provided by the authors covered this phenomenon?

See also:

Strandell, A., Hall, C.M. 2015. Impact of the residential environment on second home use in Finland – Testing the compensation hypothesis. Landscape and Urban Planning 133: 12–23. https://doi.org/10.1016/j.landurbplan.2014.09.011

Methodology. The authors could shortly discuss the complementarity or differences in carbon and energy footprint in their discussion / conclusion. Are these indicators according to their results interchangeable and in which circumstances not, what kind of recommendation do they make to forthcoming studies on the effects of household consumption? The energy portfolio varies across countries. E.g. the carbon footprint of countries having a high share of fossil fuels (coal, oil shale) versus nuclear or renewable energy should also result in different energy and carbon footprints of household consumption. There have been significant changes in energy portfolio in recent years that are not yet visible in the 2010 data used in the study: the authors should address this in their 1) methodology and 2) policy section.

As this is a comparative study covering a number of countries, the authors could acknowledge the main limitations of the international EXIOBASE database and discuss the potential effects on their results.

Author Response

Reviewer 1

This is a comprehensive and interesting comparative study on the interactions of household size and the climate related environmental effects of household consumption. The spatial coverage of all contemporary member states of the EU clearly fills a comparative gap in the field of environmentally extended household consumption studies. The authors have well related their results to broader social trends and cultural contexts. The text is well written and easy to follow, and I enjoyed reading the manuscript. Despite of a strong basis, there is one major and some minor issues to consider.

Thank you. We address your feedback in detail below.

The main consideration is related to the applied spatial scales and the resulting claims. The authors should seriously consider and conceptualise the spatial unit of exploring the economies of scale in different spatial contexts. Their current approach linking the average density of an undefined area with the potential of sharing between households (i.e. on close community and neighbourhood level) is simplistic and problematic due to the serious mismatch of spatial scales.

Although the authors claim that the methodology of household consumption survey is described elsewhere, they should clearly indicate how the density variable has been originally collected or assigned, and which spatial unit it represents (this is not evident from their given Eurostat HBS reference either). E.g. municipality average density is a very different proxy from the scale of neighbourhood density or even building type where the sharing between households may occur. In addition, the authors ignore settlement size, hierarchies and the related economies of scale that shape the contexts of opportunities to share (incl. the presence of various shared mobility schemes). This is related to the core distinction “urban-rural” of the dataset itself, which is a strong simplification, missing the complex and interlinked functionality, and population distribution of regions. If the authors miss relevant spatial data, they should be very cautious with their related claims. PS Section 3.3. and the supplementary material could depict the differences between consumption categories across spatial scales.

See further on the choice of spatial scales or understanding shared mobilities:

Burgalassi, D, Luzatti, T. 2015. Urban spatial structure and environmental emissions: A survey of the literature and some empirical evidence for Italian NUTS 3 regions. Cities 49: 134–148, https://doi.org/10.1016/j.cities.2015.07.008

Freundendal-Pedersen, M., Kesselring, S. 2018. Sharing mobilities. Some propaedeutic considerations. Applied Mobilities 3: 1–7. https://doi.org/10.1080/23800127.2018.1438235

Jones, C., Kammen, D.M. 2013. Spatial distribution of U.S. household carbon footprints reveals suburbanization undermines greenhouse gas benefits of urban population density. Environmental Science & Technology 48: 895–902. https://doi.org/10.1021/es4034364

Poom, A., Ahas, R. 2016. How Does the Environmental Load of Household Consumption Depend on Residential Location? Sustainability 8: 799. https://doi:10.3390/su8090799

Thank you very much for discussing this issue. We have made some clarifications in text related to the population density and its origin - please see below.

The degree of urbanisation classification in Eurostat [1] measures population density based on Local Administrative Unites level 2 (LAU2). LAU2 are low level administrative divisions of a country below that of a province, region and state used for the purpose of providing statistics at a local level[2]. The lower LAU level (LAU level 2, formerly NUTS level 5) consisted of municipalities or equivalent units in the 28 EU Member States[3]. The degree of urbanisation measure defined by Eurostat classifies LAU2 into sparsely, intermediate and densely populated areas, using as a criterion the geographical contiguity in combination with the share of local population living in the different types of areas[1]. A map of the degree of urbanisation in 2011 for all of the EU and a detailed explanation of the undertaken steps to the classification of LAU2 can be found elsewhere [1].

Here is some more detail on the calculation of population density in the HBSs. In a first step, the typology classifies grid cells of 1 km2 to one of the three following clusters, according to population size and density:

  • High-density cluster/urban centre: contiguous grid cells of 1 km2 with a density of at least 1 500 inhabitants per km2 and a minimum population of 50 000
  • Urban cluster: cluster of contiguous grid cells of 1 km2 with a density of at least 300 inhabitants per km2 and a minimum population of 5 000
  • Rural grid cell: grid cell outside high-density clusters and urban clusters.

In a second step, local administrative units (LAU2) are then classified to one of three type of areas:

  • Densely populated area (alternative names: cities or large urban area): at least 50 % lives in high-density clusters; in addition, each high-density cluster should have at least 75 % of its population in densely-populated LAU2s; this also ensures that all high-density clusters are represented by at least one densely-populated LAU2, even when this cluster represents less than 50 % of the population of that LAU2;
  • Intermediate density area (alternative name: towns and suburbs or small urban area): less than 50 % of the population lives in rural grid cells and less than 50 % live in high-density clusters;
  • Thinly-populated area (alternative name: rural area): more than 50 % of the population lives in rural grid cells.

Figure 1 below shows the degree of urbanisation classification by grid cells and LAU2 units, while Figure 2 demonstrates the example of the types of clusters and degree of urbanisation in the city of Cork, Ireland. (for the figures please see the uploaded word-file)

While LAU2 is the lowest available disaggregation of local administrative units that is harmonised across EU countries, we agree with the reviewer that between-household sharing potential likely depends on other factors beyond the population density, e.g. social, infrastructural and institutional factors. We do not ignore the factors of settlement size, opportunities to share on the neighbourhood or building type level, and availabilities of sharing schemes – but the Eurostat survey does not offer such detail and we hence we cannot capture their importance in our analysis. We completely agree with the reviewer that this is a simplification of the evaluation of sharing potential and extended the text to discuss these limitations.

In response to this feedback, we made changes to the text throughout to emphasize that limitation and discuss urban economies of scale as a proxy rather than a direct measure of between-household sharing opportunities. Indeed denser urban environments hold more potential compared to rural ones due to the proximity of households – allowing for shared infrastructure, use and ownership of goods. We made sure to also discuss our inability to capture neighbourhood- and dwelling-level effects, sharing initiatives etc. We checked the suggested reading and incorporated it in the text as well.

It is perhaps worth noting that the focus of our analysis is the household size effect on carbon and energy footprints and how it varies with population density rather than the effect of population density itself. Therefore, we consider that a systematic (even though relatively crude) measure of population density fits the purpose.

Household type and structure. The paper would benefit from a more structured approach to household type and structure. As consumption demand and pattern is age group specific, the authors could provide some discussion (if it is out of the scope of the analysis) on how the household structure affects the economies of scale. E.g. does the effect of adding another household member differ when (s)he is from an older, third generation or a newborn baby? Could there be a difference in one- and two-parent families?

We included more discussion on the choice of variables in the model and additional discussion on excluded variables and their potential effects, such as household composition, dwelling characteristics etc.

Transportation. The results are very interesting and as they work against normal understanding of the benefits of shared commodities, the reasoning should be stronger. E.g., could the positive association between household size and footprint lean among others on the phenomenon that an additional household member induces a need for another household car with respective costs.

See also: Ritter, N., Vance, C. 2013. Do fewer people mean fewer cars? Population decline and car ownership in Germany. Transportation Research Part A: Policy and Practice 50: 74–85. https://doi.org/10.1016/j.tra.2013.01.035

We agree with the reviewer that this is an interesting and important result to expand on. We included the suggested reference and reflected on the sub-categorical results in the transport domains. Indeed we believe that the need for another car is the reason behind the lack of substantial economies of scale with regards to car travel, and also potential offsetting effects such as driving further/having a larger car in one-car households. Also we don’t expect household economies of scale with regards to air travel or public transport.

Housing. In the contemporary world (and Europe), people are more mobile either on personal or work-related reasons and they might have several homes, including in transnational situations. On the other hand, in several European countries, there is a tradition of having second homes (summer cottage, childhood home without permanent residents). How do the HBS dataset and the analysis provided by the authors covered this phenomenon?

See also: Strandell, A., Hall, C.M. 2015. Impact of the residential environment on second home use in Finland – Testing the compensation hypothesis. Landscape and Urban Planning 133: 12–23. https://doi.org/10.1016/j.landurbplan.2014.09.011

HBS are designed to capture household expenditure through continuous consumption diaries and through additional household surveys designed to cover large infrequent expenses. While individual surveys may include questions about second homes (e.g. see the reference from the Living costs and food survey carried out in the UK (the UK HBS), expenditure associated with second homes are not a separate item in the harmonized dataset distributed by Eurostat.

http://doc.ukdataservice.ac.uk/doc/8351/mrdoc/pdf/8351_volume_b_household_questionnaire_201617_final.pdf

The additional household survey is designed to capture infrequent purchases, such as buying/improving/extending a second home, personal vehicle etc – although there may be substantial recalling and measurement inconsistencies in the accounting of these expenses. Other expenditure associated with second homes are likely not well captured in the surveys. We expanded the limitation section to capture this.

Methodology. The authors could shortly discuss the complementarity or differences in carbon and energy footprint in their discussion / conclusion. Are these indicators according to their results interchangeable and in which circumstances not, what kind of recommendation do they make to forthcoming studies on the effects of household consumption? The energy portfolio varies across countries. E.g. the carbon footprint of countries having a high share of fossil fuels (coal, oil shale) versus nuclear or renewable energy should also result in different energy and carbon footprints of household consumption. There have been significant changes in energy portfolio in recent years that are not yet visible in the 2010 data used in the study: the authors should address this in their 1) methodology and 2) policy section.

We thank the reviewer for bringing our attention to this point. We now include a discussion about this in the methodology and discussion. We find that similar trends regarding the effect of household size are present for both carbon and energy footprints (negative and significant coefficients for both carbon and energy). Yet, there are differences in the magnitude of coefficients with negative HHSIZE coefficients of larger absolute value for energy. This suggests that while we note a decrease in consumption with rising household size, we also note an increase in the carbon intensity of consumption. This increase in carbon intensity of energy use offsets some of the reductions associated with lower energy use and consumption per capita of larger households.

As we carry out the regression within countries, we do not expect that cross-country differences in the energy mix of countries will explain the differences between the HHSIZE coefficients for carbon and energy. The differences in coefficients between carbon and energy result from differences in the consumption between household cohorts of various size (e.g. smaller households consuming less of high carbon items compared to larger households per capita).

As this is a comparative study covering a number of countries, the authors could acknowledge the main limitations of the international EXIOBASE database and discuss the potential effects on their results.

We added some additional discussion about that in the Limitations section. The full list includes limited product and spatial detail, monetary input-output tables, difficulties around land use change emissions. We also discussed the limitations of focusing solely on household expenditure.

Thank you for your feedback! We hope to have addressed it in a satisfactory manner.

Reviewer 2 Report

Dear Authors,

Thank you for sharing your work. I was really keen to read how Household Sharing for Carbon and Energy Reductions: the Case of EU Countries. Although the topic is very interesting there are several major insufficiencies.

Suggestions for improvement:

- no reference to literature [53] in the text of the article-line 720

-link to literature [18] is incorrect -line 651

- are there more up-to-date literature items than from 1996 [42] -line 699

- please enter the lietarura in accordance with the requirements of the journal [21; 24] -line 658; 666

- please provide detailed information regarding item [26], which is the magazine -line 670

- how were the variables selected for the model? -line 167

-What other variables were taken into account when constructing the model?

- whether the REGION variable includes climate related correction, if so how -line 195-197

-figures 1 and 2 are difficult to read, legend is missing -line 257-261

- figure 1 is missing the EU average, for reference - line 253-255

- please explain this request - line 277-284

- do the authors include in the research the indicator of the amount of food waste? - line 402-415

- no descriptions regarding DK, PL, SE in first cluster and IT in second cluster

-discussion and conclusions requires significant improvement - the conclusions do not result directly from the data and model in the article

Author Response

Reviewer 2:

Dear Authors,

Thank you for sharing your work. I was really keen to read how Household Sharing for Carbon and Energy Reductions: the Case of EU Countries. Although the topic is very interesting there are several major insufficiencies.

Thank you for your feedback. We addressed the comments below.

Suggestions for improvement:

- no reference to literature [53] in the text of the article-line 720

References 53 and 54 are in text as footnotes on pages 6 and 15.

-link to literature [18] is incorrect -line 651

We tested the link on the February 14th and it worked fine:

https://www.un.org/development/desa/en/news/population/2018-revision-of-world-urbanization-prospects.html

- are there more up-to-date literature items than from 1996 [42] -line 699

We included more recent references pointing to traditional gender roles and family structures in Southern European countries.

- please enter the lietarura in accordance with the requirements of the journal [21; 24] -line 658; 666

We updated the references and added links and access dates:

Eurostat EU Quality report of the Household Budget Surveys 2010 Available online: http://epp.eurostat.ec.europa.eu/portal/page/portal/eurostat/home (accessed on Jan 27, 2020).

Schenau, S. The Dutch energy accounts Available online: https://unstats.un.org/unsd/envaccounting/londongroup/meeting10/LG10_10a.pdf (accessed on Jan 27, 2020).

- please provide detailed information regarding item [26], which is the magazine -line 670

This is unpublished work and we followed the following structure suggested in the authors guidelines of the journal:

  1. Author 1, A.B.; Author 2, C. Title of Unpublished Work. status (unpublished; manuscript in preparation).
  2. Ivanova, D.; Wood, R. The unequal distribution of household carbon footprints in Europe and its link to sustainability. In Review in Glob. Sustain. 2020.

- how were the variables selected for the model? -line 167

We expanded on that added some reasoning and references.

-What other variables were taken into account when constructing the model?

We based the carbon and energy footprint calculations based on a set on expenditure variables from the surveys.

- whether the REGION variable includes climate related correction, if so how -line 195-197

The REGION variable capture geographical and climatic differences indirectly. Since our regressions are conducted across countries, we believe this will generally capture climatic differences well. We expanded the text to capture this.

-figures 1 and 2 are difficult to read, legend is missing -line 257-261

- figure 1 is missing the EU average, for reference - line 253-255

Figure 1: We increased the size of the labels and added measurement in the hope to make it easier to read. The colours and symbols are not meant to be given interpretation, but they are used to enable easier reading of results where symbols/names are overlapping. We added the EU average as a reference in the old Figure 1 (now Figure 2).

Figure 2 (now Figure 3) – We added axis measures and a legend to ease interpretation. The same colour, symbol and country coding applies as in the rest of the figures.

- please explain this request - line 277-284

We are not sure what the reviewer means as “request” here. Can you please clarify?

If the reviewer wonders why we focus on these two countries, we discuss some potential explanations for why Denmark and Malta may not share the association between household size and population density that other EU countries depict.

- do the authors include in the research the indicator of the amount of food waste? - line 402-415

No, we cannot test the potential change in food waste ourselves. Through the Household Budget Survey, we only had access to food consumption of households. As we found evidence for the variation in food waste among different household sizes in prior literature, we included the reference but we were unable to test it ourselves. We added a comment on that in text.

- no descriptions regarding DK, PL, SE in first cluster and IT in second cluster

Results about these countries have also been added to the summary table.

-discussion and conclusions requires significant improvement - the conclusions do not result directly from the data and model in the article

We added more reflections on own results in the discussion and conclusions.

Thank you for your feedback! We hope to have addressed it in a satisfactory manner.

Round 2

Reviewer 1 Report

Thank you for your thorough elaboration on various methodological aspects, limitations, and discussion. The paper reads better now and the conclusions are justified. The paper is suitable for publication. 

Reviewer 2 Report

All suggestions were accepted by the authors